# The Impact of Digital Financial Inclusion and Bank Competition on Bank Stability in Sub-Saharan Africa

Tough Chinoda * and Forget Mingiri Kapingura

Department of Economics, University of Fort Hare, Private Bag X9083, East London 5200, South Africa
* Correspondence: toughchinoda@gmail.com

**Abstract:** The last few years have witnessed a rapid development in digital finance that may threaten the manner in which traditional financial services are being used. It opens up new opportunities for low-income groups and small businesses that have limited or no access to formal financial services. Thus, digital financial inclusion plays a vital role in boosting a country's financial inclusion, fulfilling some sustainable development goals and achieving higher economic growth. This study builds on a new measure of digital financial inclusion to examine the impact of digital financial inclusion and bank competition on bank stability in Sub-Saharan Africa for the period 2014 to 2020 using the two-step System Generalised Method of Moments. An index of digital financial inclusion, z-score, Herfindahl–Hirschman Index (HHI), and non-performing loans were used as data variables. The study findings reveal that digital financial inclusion has a significant positive relationship with bank stability (z-score) and a negative relationship with non-performing loans. The study also found a significant negative effect of bank competition (HHI) on bank stability in line with the competition-fragility view. Policymakers should ensure digital financial literacy for all since it feeds into bank stability and also reduces bank insolvency. They should also find ways of enhancing bank competition which reduces non-performing loans and bank insolvency. On practical implications, the study calls for strategic measures to preserve bank stability, such as complementing digital financial inclusion with financial literacy and enhancing bank competition.

**Keywords:** bank stability; bank competition; digital financial inclusion; Sub-Saharan Africa

## 1. Introduction

Digitalisation has transformed financial systems in developed and developing countries (Wysokińska 2021). Barriers in traditional financial systems continue to fall (Kooli et al. 2022), resulting in a rise in financial inclusion, which is also recognised as a key enabler for achieving the 2030 Sustainable Development Goals (SDGs) (Allen et al. 2016). As a result, digital financial inclusion has become a major topic of debate both at home and abroad. The topic of digital financial inclusion takes on special significance as global financial inclusion gaps exist due to various barriers (Demirgüç-Kunt et al. 2018; Pazarbasioglu et al. 2020). According to the World Bank Global Findex report, only 33% of the adult population owned a bank account at a formal financial institution in Sub-Saharan Africa (SSA) in 2017 which is less than any other region global (Demirgüç-Kunt et al. 2018). On the other hand, 65% of the adult population in the poorest developing countries still lack access to a formal bank account, and only 20% use a formal financial institution to save (Pazarbasioglu et al. 2020). Considering the prospects and importance of digital financial inclusion, banks in SSA countries have started implementing digital financial services in a full-fledged manner. This is because broader digital financial inclusion helps banks achieve financial advancement (Demirgüç-Kunt et al. 2018), stability (Ahamed and Mallick 2019), and a flourishing global financial sector. Globally, the banking sector is in transition, and banks, particularly those in developing nations, face fierce competition from foreign banks. These nations also have a higher percentage of unbanked people than industrialised nations

do (Kim et al. 2018). In the quest to reduce the number of the unbanked population, banks in emerging economies are resorting to the use of digital financial services so as to increase financial inclusion.

As displayed in Table A1, digital financial inclusion variables such as mobile subscription per 100 people, registered mobile money agents per 100,000 adults, active mobile money accounts per 1000 adults, and individuals using the Internet (% of the population) have increased over the period 2014 to 2020. By 2019, there were 469 million active mobile money accounts in Sub-Saharan Africa, with over 181 million accounts registered. This is similar to having a mobile money account with more than 15% active users in over 40% of the region's total population (Munoz et al. 2022). The COVID-19 epidemic caused these activities to pick up speed. Mobile money-driven activities increased along with the digital economy, leading to notable developments in some African nations, including Zimbabwe (Ecocash), Kenya (M-Pesa), and South Africa (E-wallet), among others. This justifies the need for crafting policies that enhances digital financial inclusion and bank competition in the region. However, it has also been contended that competition and digital finance have their related challenges that could result in financial instability (Nelson 2018, 2019; Ozili 2018; Pazarbasioglu et al. 2020). Bank stability in the region seems to have been threatened over the period as witnessed by an increase in non-performing loans (NPLs) from 6.5% in 2014 to 12.72% in 2020. The rise in non-performing loans can hamper banks' ability to increase financial intermediation. High and rising levels of non-performing loans can put a strain on bank balance sheets, affecting lending operations (Laib and Abadli 2018), and limiting banks' ability to increase financial intermediation. The rise in NPLs can lead to disintermediation in the economy, with banks reducing lending and requiring borrowers to put up security even for minor loans. NPLs can also affect bank performance and lower the sector's profitability through increasing provisioning. Bank Z-Score, however, reflects otherwise as the values have increased over the same period.

Tandelilin and Hanafi (2021) investigated the role of institutional quality on the nexus between bank competition and stability in Asia. Employing the GMM estimator on a sample of 427 commercial banks in Asia, the study finds that competition erodes bank stability, whilst institutional quality can promote and mitigate the negative effect of bank competition on stability. These results support the findings of the study conducted by Albaity et al. (2019) on 276 banks in the Middle East and North America for the period 2006 to 2015. The study concluded that banks facing more competition tended to take more credit risks which reduces profitability in line with the competition-fragility view.

For West Africa, Cobbinah et al. (2020) employed the Generalised Method of Moments and Panel Vector Autoregressive estimation to examine the nexus between banking competition and bank stability in West Africa for the period 2000 to 2014. They concluded a significant positive effect of banking competition on bank stability in line with the "competition-stability view". These results support the findings of the study conducted by Ijaz et al. (2020) on European countries between 2001 and 2017. The results of the study reveal that lower banking competition boosts economic growth and increases financial stability. The above results on the nexus between bank competition and bank stability are conflicting and warrant further investigations. In addition, these studies, however, did not consider the effect of digital financial inclusion, which affects bank competition. Our study seeks to close the gap using the current time frame and dataset.

Studies on the nexus between digital financial inclusion and bank stability are limited, warranting further research. Ahamed and Mallick (2019) investigated the impact of traditional financial inclusion but not digital financial inclusion on bank stability in different international samples and concluded a positive effect of financial inclusion on bank stability. In addition, Danisman and Tarazi (2020) and Boachie et al. (2021) investigated the interplay between financial inclusion and banking stability in Europe and Sub-Saharan African countries, respectively. The results of the studies suggest that financial inclusion positively and significantly influences bank stability in these regions. These results are also in line with Jungo et al. (2022), who employed the Feasible Generalised Least Squares (FGLS) model to

assess the moderating role of financial regulation on the effect of financial inclusion and competitiveness on banks' financial stability in SSA and Latin American and Caribbean (LAC) countries. The results suggest a positive effect of financial inclusion on bank stability in SSA and LAC countries, and financial regulation in LAC countries increases financial stability. The study also suggests an inverse relationship between competitiveness and financial stability in SSA and LAC countries. However, these studies have been criticised for failing to comprehensively define financial inclusion as they excluded digital financial inclusion variables and only used traditional financial inclusion variables. Our study seeks to close this gap by including digital financial inclusion variables.

Banna (2020) and Banna and Alam (2021a, 2021b) examined the role of digital financial inclusion in promoting sustainable economic growth through banking stability in Bangladesh, Islamic regions, and emerging Asian countries by deploying panel-corrected standard errors and two-stage least square-instrument variable methods. The results of the studies reveal that digital financial inclusion accelerates banking stability, which not only reduces bank default risk but also increases financial mobility in the regions. In another study, Banna and Alam (2021c) examine the nexus between digital financial inclusion and bank risk-taking levels, using a sample of 283 commercial banks from six countries over the period 2011 to 2019. Deploying two-stage least squares-instrumental variables, panel-corrected standard errors, and dynamic panel two-step generalised method of moments estimators, the study found that an increase in the digital financial inclusion index score reduces the overall bank risk-taking level and increases banking stability for conventional and commercial banks compared to Islamic banks. Banna (2020) also used principal component analysis, a two-step dynamic system generalised method of moment analysis, ordinary least square, and panel-corrected standard error approaches to investigate the nexus between digital financial inclusion and bank stability in ASEAN countries. The empirical study finds that digital financial inclusion accelerates ASEAN banking stability, which not only reduces bank default risk but also increases financial mobility in the region, using an unbalanced panel data of 213 banks from 4 ASEAN nations. The findings also imply that by implementing digital financial inclusion, ASEAN banks will be able to maintain banking sector stability by decreasing liquidity crises and non-performing loans in the post-COVID-19 future. Banna et al. (2021) deployed the Panel-Corrected Standard Errors, Two-Stage Panel Least Squares-Instrumental Variables and Two-Step System Generalized Method of Moments dynamic panel estimation method to investigate the nexus between digital financial inclusion and bank stability using an unbalanced panel data of 65 Islamic banks from six countries between 2011 and 2020. The result suggests that digital financial inclusion accelerates Islamic banking stability in the studied region. None of these studies were done in SSA, where there are dynamic geographic, economic, and social characteristics in terms of economic growth, natural resources, levels of informality, economic development, population growth, rates of poverty, and unemployment (Jungo et al. 2021). Banking systems in Islamic and ASEAN regions are different from SSA, thus justifying our study.

It is clear from the above studies that there is a dearth in the literature that specifically focuses on the relationship between digital financial inclusion, bank competition and bank stability in Sub-Saharan Africa. The concept of digital financial inclusion has become the catchphrase in the field of financial economics research, but its link with bank competition and bank stability is one novel contribution of this paper. Constructed on the debate that digital financial inclusion is an essential avenue and a game changer in ascertaining progress towards the accomplishment of the United Nations Sustainable Development Goals (SDGs) through global financial inclusion, particularly in developing countries, this paper focuses on digital financial inclusion, bank competition and bank stability in Sub-Saharan Africa where a considerable number of adults remain unbanked. Specifically, the analysis of the combined impact of digital financial inclusion and bank competition on bank stability is the first of its kind to the best of our knowledge. Secondly, we computed an index of digital financial inclusion in the region. The implication is that competition, and digital financial inclusion are significant in explaining the stability of the banking

environment. These findings reveal some grey areas requiring attention when crafting policies to stimulate stability. The rest of the paper is organised as follows. Section 2 elucidates the methodology of the study, while Section 3 entails results. Discussion and conclusion with policy recommendations are illustrated in Sections 4 and 5, respectively.

## 2. Methodology

### 2.1. Econometric Model

Resultantly, the nexus between digital financial inclusion, bank competition and bank stability is assumed to take the following general form of linear dynamic panel model:

$$\aleph_{i,t} = \omega_i + \rho_i \, \aleph_{i,t-1} + X_{i,t} \, \rho + \varepsilon_{i,t} \tag{1}$$

where: $t = r + 1 \ldots T$, and $\varepsilon_{i,t}$ is assumed to be serially uncorrelated. The estimating regression model is derived from Equation (1) and is written as:

$$\aleph_{i,t} = \omega_i + \beta_{k,it} \, \varphi_{k,it} + \beth_{k,it} + \Omega_{k,it} + \varepsilon_{i,t} \tag{2}$$

where: $\aleph$ represents bank stability proxied by non-performing loans (NPLs) and bank Z-Score (LNZSCORE) respectively, k indicates the regressions number, it represents country i in time t, $\beta$ stands for digital financial inclusion; $\beth$ stands for economic development represented by growth in gross domestic product, $\Omega_{k,it}$ stands for bank competition and $\varepsilon_{i,t}$ is the error term.

### 2.2. Variables and Description

#### 2.2.1. Bank Stability

This study used non- performing loans as a percentage of total loans and bank Z-Scores to measure bank stability following Kim et al. (2020) and Syed et al. (2021), respectively. A high non-performing loan ratio denotes bank instability due to high default risk. The study expected digital financial inclusion to reduce non-performing loans, thus positively affecting bank stability. Literature has widely used the Z-Score as an unbiased bank riskiness indicator (Banna et al. 2021). The Z-Score reflects the probability of bank failure. This measure portrays a banking system buffer (return and capitalisation) to the standard deviation in returns (Klapper and Lusardi 2020). It also reflects the number of volatility (standard deviations) where profits may decline before a bank fails (Klapper and Lusardi 2020). The higher the bank's Z-Score, the lower the chances of bank insolvency. To reduce data skewness, we followed Ahamed and Mallick (2019) and used the natural logarithm of Z-Score.

#### 2.2.2. Digital Financial Inclusion

Since this study seeks to analyse the impact of digital financial inclusion and bank competition on bank stability in Africa. Data from the World Development Indicators Database (World Bank) has been used to measure digital financial proxies for the period 2014 to 2020. The panel data was balanced for all selected countries. This period was selected based on data availability for the employed variables, providing the rudimentary practical policy implications and addressing the research questions. The study considered both the digital financial usage and outreach penetrations in computing digital financials based on previous studies (e.g., Ahamed and Mallick 2019; Banna and Alam 2021a, 2021b, 2021c). Following Khera et al. (2021), we constructed an enhanced digital financial inclusion index which is different from the traditional financial inclusion index. Unlike Khera et al. (2021), we added indicators of the use of digital financial transactions and access to physical infrastructure. The indices consist of access and usage indicators provided by digital financial services, including fintech companies, mobile money operators, and other new entrants in the financial sector. We used the modern multivariate data analysis tool of a three-stage principal component analysis (PCA) to construct "usage" and "access" sub-indices, to capture supply and demand-side aspects of financial inclusion, respectively. The PCA is also used to combine access and usage sub-indices into a digital financial

inclusion index, to capture financial inclusion through financial institutions and enabled by technology separately. The PCA technique retains all available data variations, reduces data dimensionality, and resolves the possible multicollinearity that may arise among the variables (Nizam et al. 2020). Using PCA, each dimensional indicator is normalised to have values between zero (0) and one (1) to make immaterial the scale that they were measured. Afterwards, the PCA extracts the common principal component of the dimensions that capture various aspects of digital finance. As usage variables, we considered the number of active mobile money accounts per 1000 adults. We also considered the % population with access to the Internet and mobile subscription per 100 people as indicators for access to digital infrastructure and the number of registered mobile money agents per 100,000 adults as availability. We developed an index of DFI using principal component analysis.

### 2.2.3. Bank Competition

Several studies have used the Lerners index and the Boone indicator to proxy bank competition. There are several missing observations on the Boone indicator and Lerner index, thus reducing the sample size. The CR4 ratio shows the largest 4 banks' percentage market share and is an easy way of visualising market asset control. This measure has a disadvantage in that it ignores the small banks. We used the HH index, which builds on this drawback by capturing features of the entire distribution of bank sizes.

### 2.3. *Estimation Technique*

System Generalised Method of Moments

Different approaches have been utilised in the literature to investigate the impact of digital financial inclusion and bank completion on bank stability. We followed Akande et al. (2018) in employing the Arellano and Bond (1991) and Arellano and Bover (1995) dynamic versions of the System Generalised Method of Moments (S-GMM) to explore the relationship in the SSA region. The adoption of this approach was backed by several motivations, which include the need to account for probable endogeneity and the handling of cross-sectional dependence incidences since the method is presumed to be identically and independently distributed. In addition, the increasing usage of the panel data technique, which allows individual cross-section dynamics in economic and finance studies, has favoured the application of dynamic panel data analysis. Moreover, the conventional dynamic panel data estimators such as first difference, pooled Ordinary Least Squares and Generalised Least Squares, among others, have been criticised for failing to deal with the inclusion of lagged endogenous variables in a model with individual effects which the S-GMM can efficiently handle. In addition, S-GMM is a normality-free regression technique with great adaptability and data-generating process assumptions, with dependent variables being instrumented by their lagged variables. Panel data analysis alleviates multicollinearity issues among variables, allows the removal of unobserved heterogeneity for each sample observation (Arellano and Bover 1995), accommodates the creation and analysis of more difficult behavioural models, and assists in studying countries in the SSA region that would not have been studied due to inadequate information. For the validity of the S-GMM instruments, we used the Hansen J statistics in robust estimation.

Because lagged values of regressors are poor instruments for the GMM equation in difference form and difference equations can suffer from small sample bias, this study uses two-step system estimators to obtain efficient and consistent approximations of parameters. The consistency of the GMM estimators developed by Arellano and Bover (1995) is based on a set of assumptions about the error term. These assumptions imply that the error term is not serially linked with instruments, making it suitable for inclusion in the instruments matrix. This study uses a series of specification tests to test these assumptions. The first test examines whether the error term exhibits serial correlation at the second order. Due to the specification of the equations of the GMM estimator, first-order serial correlation may be present in differenced residuals even if it is not present in the original residuals. As a result, the goal of this research is to keep the null hypothesis for second-order serial correlation

from being rejected. The second test analyses if the limits have been over-identified. The results show that the instruments are holistically valid, as well as the GMM estimator's moment conditions. This study employs the robust command in Stata to modify standard errors in two-step estimation that are strongly downward biased.

*2.4. Data Sources*

The study considers 22 African countries (see Appendix A) for the period 2014 to 2020, whose main agenda is to perform their financial transactions in a cashless manner, enhancing digital financial inclusion. These 22 countries can be classified into three subgroups, namely low-income countries (LIC), lower-middle-income countries (LMIC), and upper-middle-income countries (UMIC) under the 2020 World Bank gross national income (GNI) per capita (World Bank 2020). The selection of these countries and the period under study was also based on data availability for the employed variables, providing the rudimentary practical policy implications and addressing the research questions. Data were extracted from the International Monetary Fund, Global Findex databases for DFI data, and the World Development Indicators, World Bank database.

*2.5. Robustness Check*

The analysis of the data used in this study was also carried out using static (pooled Ordinary Least Squares-fixed effect and random effect) models, and the results are presented in Table A4. The essence was to help validate the results of the GMM model and to provide evidence to substantiate the results' robustness even in the long term. The focus of this analysis was on the GMM, although the static models were also provided for further robustness purposes. The GMM results based on the robust corrected standard error provide results that are, in most cases, consistent with some static results. For instance, DFI is shown to relate positively to bank stability (lnZ-score) (and consistent with the GMM models), implying that an increase in digital financial inclusion will increase bank stability. An increase in bank competition (HHI) will reduce bank stability (lnZ-score) in line with the competition-fragility view under the static model and with signs consistent with the GMM model, although the effect is not significant for the static model. Similarly, economic development (lnGDPGR) reduces NPLs under the static and GMM model and is significant apart from the static model, where the effect is not significant, as indicated in Table A4.

## 3. Results

*3.1. Summary Statistics*

The summary statistics of digital financial inclusion, bank competition, and bank stability in SSA are illustrated in Table A2. Digital financial inclusion in SSA has a mean of 31.4% with minimum and maximum values of 3% and 71%, respectively, which implies serious digital financial inclusion discrepancies in Africa, consistent with Thaddeus et al. (2020). The standard deviation of 16% suggests low levels of digital financial inclusion as the mean (31.4%), and volatility fails to exceed the 50% level. This reveals that the digital economy and mobile money-driven activities have expanded in SSA, resulting in noteworthy outcomes in certain African countries such as Zimbabwe (Ecocash), Kenya (M-Pesa), and South Africa (E-wallet). However, other countries like Comoros and Madagascar still lag behind. The discrepancies could be a result of a lack of adequate cyber security and data protection laws, norms, and regulations which may breed mistrust and undermine effective digital financial inclusion. In 2020, cyber security was ranked among the top five threats by 76.1% of respondents, according to the 2020 Global Risks Report by World Economic Forum (2020). Economic development was at 4% on average, ranging between −10.78% and 10.82%. Economic development in SSA is positive, despite countries like the Congo Republic having the least development of −10.78% in 2016. Bank stability (z-score) in SSA averages 15.82% reflecting that banks in the region are less stable. Bank stability measured by NPLs reveals a mean of 10.1% and maximum and minimum values of 60.05% and 1.45%, respectively. The bank competition (HHI) has a mean of 13.8% with a maximum and minimum of 40%

and 3%, respectively. The standard deviation of 9.2% suggests a low concentrated banking system in SSA as the mean is fundamentally different from the maximum value. The bank market in the region operates in a monopolistic market.

### 3.2. Correlation Analysis

We also estimated the correlation between the variables, and the results are indicated in Table A3. The study found a significant positive association between digital financial inclusion and bank stability (ZSCORE), providing evidence that digital financial inclusion is favourable for bank stability measured by ZSCORE and economic development. On the other hand, the study found a significant negative association between digital financial inclusion and variables such as bank competition (HHI) and bank stability (NPLs), providing evidence that bank competition and non-performing loans (NPLs) hurt digital financial inclusion in line with the information hypothesis. This makes perfect economic sense since it means advancement in digital financial inclusion in SSA helps in maintaining banking stability by lowering NPLs since borrowers can easily use digital platforms instead of travelling, and they are also cheap to use. This assists in creating a more stable banking system. The results are in line with Banna et al. (2021), and Risman et al. (2021). Bernini and Montagnoli (2017) evidenced that competition enhances the pressure of financial needs by firms but that the firms get discouraged from applying for loans as a result of complex loan approval processes and high loan costs. Ayalew and Zhang (2019) admit that several firms in Africa get discouraged from making loan applications irrespective of the bank's high loan application approval rates. The correlation between NPLs and bank stability (z-score) is negative, supporting that NPLs hurts bank stability. On the other hand, the association between NPLs and bank competition is positive. Bank competition and bank stability are negatively associated, providing evidence that bank competition hurts bank stability. The inverse relationship between bank competition and bank stability is in line with the competition fragility view, which contends that competition enhances banks' incentives to risk-taking, which thus threatens the system's stability. The correlation between economic development and variables such as digital financial inclusion and bank stability (z-score) is positive, implying that economic development supports digital financial inclusion and bank stability and vice versa. On the other hand, the association between economic development and bank competition is negative. Overall, the results suggest that there are no multicollinearity issues among the estimation variables.

### 3.3. Digital Financial Inclusion, Bank Competition and Bank Stability

The results of the Arellano and Bond two-step system GMM estimator on the impact of digital financial inclusion and bank competition on bank stability measured by NPLs and LNZSCORE are presented in Table A4. To ensure the bank's stability goal, banks should keep non-performing loans as low as possible. The study findings reveal that digital financial inclusion has a significant positive relationship with bank stability (z-score) and a negative relationship with non-performing loans. The results propose that higher levels of digital financial inclusion significantly enhance bank stability (greater stability is denoted by higher levels of z-score and reduced NPLs level in a country). DFI has allowed many people in SSA to join and participate in the financial sector, which makes the sector more stable due to the evolution of new credit facilities and other commercial activities, which permits the start of a wide range of financial products and services as financial organisations pursue a steady increase in income. Moreover, proper DFI application has increased the banks' profitability, which brings financial growth and stability. This suggests that bank soundness in various countries is enhanced through digital financial inclusion. This makes sense since digital financial inclusion in SSA can enhance bank stability through enhanced trust in the financial system. The more the number of loans and deposits digitally provided, the lower the probability that loans from financial institutions would become defaulted. The regression coefficient of digital financial inclusion and NPLs is 0.1612 and −2.4514 and significant at 10% and 5% levels, respectively, which show that a 1% increase

in digital financial inclusion reduces NPLs by 245.14% and enhances bank stability (z-score) by 16.12%. This means digitally inclusive financial sectors reduce credit risk and enhance bank stability in Africa. These findings corroborate Ahamed and Mallick (2019) and Banna and Alam (2021b), who found that financial systems with highly inclusive digital financial services tend to enhance bank stability and that greater DFI implementation reduces the NPLs of a bank.

The study also found a significant negative effect of bank competition (HHI) on bank stability (NPLs and Z-Score). A 1% increase in bank competition reduces NPLs and Z-Score measure by 93.7% and 47.2%, respectively, in line with the competition-fragility view. The study also found a significant negative effect of economic development on bank stability (NPLs). A 1% increase in economic development reduces NPLs by 31.9%, respectively. These findings, however, contradict the competition-stability view (Saha and Dutta 2020).

## 4. Discussion

The study provides empirical evidence that digital financial inclusion has a significant negative impact on NPLs and a positive effect on bank stability (lnZ-score). This implies that bank stability (lnZ-score) is boosted at a higher level of digital financial inclusion in Africa. The regression coefficient of digital financial inclusion and NPLs is 0.1612 and −2.4514 and significant at 10% and 5% levels, respectively, which shows that a 1% increase in digital financial inclusion reduces NPLs by 245.14% and enhances bank stability (lnZ-score) by 16.12%. This means digitally inclusive financial sectors reduce credit risk and enhance bank stability in Africa. These findings corroborate Banna and Alam (2021b) and Ahamed and Mallick (2019), who found that financial systems with highly inclusive digital financial services tend to enhance bank stability and that greater DFI implementation reduces the NPLs of a bank.

The study also found a significant negative effect of bank competition HHI) and economic development on financial stability (NPLs). A 1% increase in bank competition and economic development reduces NPLs by 93.7% and 31.9%, respectively. This finding is not surprising given that the increase in GDP (economic development) is usually accompanied by an increase in employment and, most likely, higher incomes over the estimated time, as a major section of the population will receive salary increases when new union agreements are negotiated. As a result, individuals had more options for obtaining and repaying new loans based on a variety of factors, including their level of salary. This can also be explained by the competition-efficiency view. Akande et al. (2018) state that competition helps make the financial sector more efficient in terms of screening and monitoring potential borrowers, thus enhancing asset quality by reducing non-performing loans, which makes banks more profitable. Moreover, banks facing greater competition earn lower interest margins and make investments with lower risks. Saha and Dutta (2020) also echoed that bank asset quality normally improves once the threat of entry increases supporting our findings. The negative relationship between bank competition and bank stability in SSA could also be a result of bank competition which erodes the pricing power of banks, thus increasing the banks' risk-taking behaviour, which is detrimental to bank stability. This is consistent with Akande and Kwenda (2017), who concluded the same results in Africa using the Lerners Index and Z-Score. Although the estimated stability measure (Z-Score) does not suggest any form of instability in the banking sectors of the SSA region, however, banks in this region continue to face high NPLs risk among others in their asset portfolios. These are likely to emanate from undue competition that makes banks in SSA not pay proper attention or wave the Know Your Customer processes and other corporate governance issues that emanate during the selection of loan assets portfolio. Bearing in mind that bank instability in SSA is associated with competition calls for caution among regulators, players, and practitioners alike so that they align their priorities to avoid such eventualities. This also makes sense as the entrance of new banks in SSA results in threats, especially when foreign banks compete with local banks affecting stability.

## 5. Summary and Conclusions

Banks play a pivotal role in the effectiveness of programs aimed at enhancing financial inclusion since they are the key providers of financial services and products in any economy (Anarfo et al. 2020; Musau et al. 2018). Banks have to be financially stable in order for them to perform their financial intermediary role efficiently (Musau et al. 2018). Therefore, in developing countries, policymakers have privileged in their agendas measures such as financial inclusion to enhance financial stability and development (Emara and El Said 2019). Inclusive economic growth and financial development must balance financial stability and financial inclusion since prioritising only one component would stifle the other (Jungo et al. 2022). This study examines the impact of digital financial inclusion and bank competition on bank stability in Sub-Saharan Africa for the period 2014 to 2020 using the two-step System Generalised Method of Moments. An index of digital financial, z-score, HHI and non-performing loans were used as data variables. The study findings reveal that digital financial inclusion has a significant positive relationship with bank stability (z-score) and a negative relationship with non-performing loans. The study also found a significant negative effect of bank competition (HHI) on bank stability in line with the competition-fragility view. Policymakers should ensure digital financial literacy for all since it feeds into bank stability and also reduces bank insolvency. They should also find ways of enhancing bank competition which reduces non-performing loans and bank insolvency. On practical implications, the study calls for strategic measures to preserve bank stability, such as complementing digital financial inclusion with financial literacy and enhancing bank competition. The proponents of the competition stability view emphasise the role of efficiency in the competition and stability nexus, which was not taken into account in this study. Future studies could also compare results in Africa against other regions to visualise the gaps that exist. The study also did not include infrastructure on the index. For instance, access to digital outlets, such as financial kiosks in convenience stores, is essential for digital financial transactions because a significant part of employees and the self-employed get wages and payments in cash.

**Author Contributions:** Conceptualisation, T.C. and F.M.K.; methodology, T.C.; software, T.C.; validation, T.C. and F.M.K.; writing—original draft preparation, T.C.; writing—review and editing, T.C. and F.M.K.; supervision, F.M.K. All authors have read and agreed to the published version of the manuscript.

**Funding:** This research received no external funding.

**Informed Consent Statement:** Not applicable.

**Data Availability Statement:** Data supporting reported results are available after request.

**Acknowledgments:** I acknowledge comments provided by participants of the Post-Doctoral Fellowship students at the University of Fort-Hare.

**Conflicts of Interest:** The authors declare no conflict of interest.

## Appendix A

**Table A1.** An Overview of Digital Financial Inclusion, Bank Competition and Stability in Sub-Saharan Africa.

| Period | 2014 | 2015 | 2016 | 2017 | 2018 | 2019 | 2020 |
|---|---|---|---|---|---|---|---|
| Non-performing loans | 6.5 | 7.36 | 9.61 | 11.24 | 11.16 | 11.34 | 12.72 |
| Z-Score | 13.59 | 15.44 | 15.94 | 16.85 | 16.30 | 16.66 | 15.96 |
| HHI | 0.127 | 0.126 | 0.129 | 0.135 | 0.151 | 0.152 | 0.147 |
| Mobile subscription per 100 people | 90 | 92 | 93 | 94 | 94 | 100 | 102 |
| Registered mobile money agents per 100,000 adults | 139 | 206 | 282 | 356 | 471 | 660 | 821 |

**Table A1.** *Cont.*

| Period | 2014 | 2015 | 2016 | 2017 | 2018 | 2019 | 2020 |
|---|---|---|---|---|---|---|---|
| Active mobile money accounts per 1000 adults | 190 | 242 | 273 | 318 | 393 | 461 | 498 |
| Individuals using the Internet (% of population) | 19 | 23 | 25 | 28 | 33 | 37 | 40 |

N/B-Data were sourced from 22 Sub-Saharan African countries, which include Botswana, Cameroon, Comoros, Congo Rep, Egypt, Ghana, Guinea, Guinea-Bissau, Lesotho, Kenya, Madagascar, Malawi, Mali, Namibia, Nigeria, Rwanda, Seychelles, Eswatini, South Africa, Uganda, Zambia.

## Appendix B

**Table A2.** Summary Statistics.

| | DFI | NPLs | HHI | ZSCORE | GDPGR |
|---|---|---|---|---|---|
| Mean | 0.314 | 10.136 | 0.138 | 15.82 | 2.400 |
| Maximum | 0.710 | 60.050 | 0.400 | 49.50 | 10.82 |
| Minimum | 0.030 | 1.453 | 0.030 | 3.00 | −10.78 |
| Std. Dev. | 0.160 | 8.360 | 0.092 | 8.87 | 4.003 |
| Probability | 0.140 | 0.000 | 0.000 | 0.232 | 0.000 |
| Observations | 154 | 154 | 154 | 154 | 154 |

*Source*: Authors Estimation.

## Appendix C

**Table A3.** Correlation Analysis.

| | DFI | NPLs | HHI | LNZSCORE | GDPGR |
|---|---|---|---|---|---|
| DFI | 1.00000 | −0.23665 | −0.14127 | 0.20888 | 0.0227 |
| NPLs | −0.23665 | 1.00000 | 0.08840 | −0.13243 | −0.0073 |
| HHI | −0.14127 | 0.08840 | 1.00000 | −0.16872 | −0.2110 |
| LNZSORE | 0.20888 | −0.13243 | −0.16872 | 1.00000 | 0.2290 |
| GDPGR | 0.0227 | −0.0073 | −0.2110 | 0.2290 | 1.0000 |

*Source*: Authors Estimation.

## Appendix D

**Table A4.** Digital Financial Inclusion, Bank Competition and Bank Stability Estimated Results.

| Variable | OLS-Random Effect (LNZSCORE) | OLS-Fixed Effect (NPLS) | 2 System-GMM (NPLs) | 2 System-GMM (LNZSCORE) |
|---|---|---|---|---|
| L.NPLs | | | 1.1739 *** (0.000) | |
| L.LNZSCORE | | | | 0.7100 *** (0.000) |
| DFI | 0.789 *** (0.001) | 13.371 *** (0.010) | −2.4514 *** (0.010) | 0.1612 ** (0.081) |
| LNGDPGR | 0.032 (0.139) | −0.723 (0.179) | −3.188 *** (0.000) | −0.003 (0.224) |
| HHI | −0.0967 (0.781) | 17.73 *** (0.028) | −9.370 *** (0.000) | −0.472 *** (0.000) |
| Constant | 2.345 *** (0.000) | 4.153 (0.105) | 1.7204 *** (0.000) | 0.913 *** (0.000) |
| AR2 | | | 0.273 | 0.20 |
| Hansen | | | 0.361 | 0.368 |
| Observations | 154 | 154 | 132 | 132 |
| Prob > chi2 | | | 0.0000 | 0.0000 |

Statistical significance ** $p < 0.1$, *** $p < 0.05$.

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
