# Peer review of "The Impact of Digital Financial Inclusion and Bank Competition on Bank Stability in Sub-Saharan Africa"

_economies, doi:10.3390/economies11010015_

Round 1

Reviewer 1 Report

I would like to thank the authors for this research that aims to aims to investigate, in the context of the Sub-Saharan Africa banking industry, how competition and digital financial inclusion promotes banking stability.

The research subject is timely, innovative, and highly interesting. It also fits the aim and scope of the journal.

The research is well designed and follows a sound scientific research method.

Results and recommendations and implications are clear and could have an impact among the community of researchers.

However, some modifications are needed in order to improve the quality of the paper.

In the last part of the research, you did not include a conclusion or you forgot to make a heading.

You need to add the limitations of the research.

There are several typos and orthograph errors (see attachment).

Other minor comments are directly attached to the manuscript.

Author Response

i have attended to the reviewers comments, see highlighted areas in yellow in the attached document. The table below also shows point by point response to the reviewer.

YELLOW: REVIEWER 1

Comment

Response to Comment

Some modifications are needed in order to improve the quality of the paper

Modifications done as highlighted in yellow in the manuscript to improve the quality of the paper

In the last part of the research, you did not include a conclusion or you forgot to make a heading.

Conclusion included

You need to add the limitations of the research.

Limitations of the research included

There are several typos and orthograph errors (see attachment).

Typos and orthograph errors attended to, some highlighted in yellow in the manuscript

Other minor comments are directly attached to the manuscript.

Minor comments attended to in the revised manuscript

Reviewer 2 Report

 The paper investigates the Impact of Digital Financial Inclusion and Bank Competition on Bank Stability in Sub-Saharan Africa.  I have read the complete paper and I hold the following observation on it.

  • The paper is well written. Although the presentation and the proofreading could be further improved. 
  • Some details around the data would be useful in the abstract.
  • The paper is short. Hence, the literature review/State of Art could be further expanded.  
  • the authors use " twenty two African countries for the period 2014 to 2020’’. It would be of interest to get a more precise idea of which typology of selected countries is involved. Is the panel balanced or unbalanced? I feel that simply taking the panel without - at least - checking if there were some inter-sectoral differences might be incomplete. More explanation on sample data is needed.
  • More explanation is needed on the definition of the selected econometrics models and the logic behind the model selection.
  • The econometric models need to be revised, there are extra spaces between the variables. Also all the variables/letters used in the models should be defined.
  • The presentation of the tables needs to be improved. Perhaps the tables could have been presented in the main text.
  • It would be important to see some robustness/sensitivity checks beyond what the authors have done. Has any robustness check been done to confirm the validity of the findings?

I wish you the best of luck.

Author Response

I have attended to the reviewers comments as shown in the table below and also as highlighted in green in the attached manuscript.

GREEN: REVIEWER 2

Comment

Revision

The presentation and the proofreading could be further improved

The presentation and the proofreading improved as highlighted in green.

·         Some details around the data would be useful in the abstract.

Details around the data included in the abstract

·         The paper is short. Hence, the literature review/State of Art could be further expanded.  

Literature review expanded slightly

It would be of interest to get a more precise idea of which typology of selected countries is involved. Is the panel balanced or unbalanced? I feel that simply taking the panel without - at least - checking if there were some inter-sectoral differences might be incomplete. More explanation on sample data is needed.

Typology of selected countries mentioned, balanced panel data used and included.

More explanation on sample data is included.

·         More explanation is needed on the definition of the selected econometrics models and the logic behind the model selection.

Explanation on the definition of the selected econometrics models and the logic behind the model selection included

The econometric models need to be revised, there are extra spaces between the variables. Also all the variables/letters used in the models should be defined

The econometric models was revised, extra spaces between the variables removed.

All the variables/letters used in the models were defined

·         The presentation of the tables needs to be improved. Perhaps the tables could have been presented in the main text.

The presentation of the tables was improved.

Presentation of Tables maintained in line with Journal requirements/ authors guidelines.

It would be important to see some robustness/sensitivity checks beyond what the authors have done. Has any robustness check been done to confirm the validity of the findings

Robustness/sensitivity checks done in section 3.4